# Acoustic signatures in Mexican cavefish populations inhabiting different caves

**Carole Hyacinthe**[1,2]ᐤ*, **Joël Attia**[3]ᐤ*, **Elisa Schutz**[3], **Lény Lego**[3], **Didier Casane**[4,5], **Sylvie Rétaux**[1]*

**1** Paris-Saclay Institute of Neuroscience, CNRS, Université Paris-Saclay, 91400, Saclay, France, **2** Department of Genetics, Harvard Medical School, Blavatnik Institute, Boston, MA, United States of America, **3** Equipe de Neuro-Ethologie Sensorielle, CRNL, CNRS and Université de St Etienne, Saint-Étienne, France, **4** Université Paris-Saclay, CNRS, IRD, UMR Évolution, Génomes, Comportement et Écologie, 91190, Gif-sur-Yvette, France, **5** Université Paris Cité, UFR Sciences du Vivant, 75013, Paris, France

ᐤ These authors contributed equally to this work.
* carole_hyacinthe@hms.harvard.edu (CH); joel.attia@univ-st-etienne.fr (JA); sylvie.retaux@cnrs.fr (SR)

**Data Availability Statement:** All relevant data are within the paper and its Supporting Information files.

**Funding:** Work supported by: a Lidex Neuro-Saclay collaborative grant to SR and JA (no website), an Equipe FRM grant (DEQ20150331745) to SR

## Abstract

Complex patterns of acoustic communication exist throughout the animal kingdom, including underwater. The river-dwelling and the Pachón cave-adapted morphotypes of the fish *Astyanax mexicanus* are soniferous and share a repertoire of sounds. Their function and significance is mostly unknown. Here, we explored whether and how sounds produced by blind cavefishes inhabiting different Mexican caves may vary. We compared "Clicks" and "Serial Clicks" produced by cavefish in six different caves distributed in three mountain ranges in Mexico. We also sampled laboratory-bred cavefish lines originating from four of these caves. Sounds were extracted and analyzed using both a manual method and a machine learning-based automation tool developed in-house. Multi-parametric analyses suggest wild cave-specific acoustic signatures, or "accents". An acoustic code also existed in laboratory cavefish lines, suggesting a genetic basis for the evolution of this trait. The variations in acoustic parameters between caves of origin did not seem related to fish phenotypes, phylogeography or ecological conditions. We propose that the evolution of such acoustic signatures would progressively lead to the differentiation of local accents that may prevent interbreeding and thus contribute to speciation.

## Introduction

Animal communication brings together all the information exchanged between individuals of the same or different species. The emitter produces a signal encoding an information, which causes a change in behavior or physiological state of the recipient [1]. In the aquatic environment, where the speed of sound propagation is approximately four times faster than in the air and travels long distances, mammalian and non-mammalian vertebrates such as teleosts largely rely on acoustic communication. In fishes, acoustic signals are mainly produced by stridulation, swim bladder pulsation, hydrodynamic movement, tendon vibration and air

(https://www.frm.org/), an Ecos-Nord exchange Program (M15A03) to SR and Patricia Ornelas-Garcia (https://www.univ-spn.fr/ecos-nord/). a Fondation des Treilles prize fellowship to CH (https://www.les-treilles.com/). The funders had no role in study design, data collection and analysis, decision to publish, or preparation of the manuscript.

**Competing interests:** The authors have declared that no competing interests exist.

release [2]. They play key roles in basic and complex behaviors such as feeding, reproduction, hierarchy, predator detection, orientation and habitat selection [3–5].

Sonic animals have their own sound repertoires. The underwater soundscape is extraordinarily diverse and representative of species such as the emblematic dolphin's clicks, baleen's songs, or toadfish's boat-whistles. The acoustic repertoire of some species can further be refined to individual signature level. Unlike voice cues that affect all calls of an animal, signature whistles in bottlenose dolphins are distinct whistle types carrying an individual's identity, as well as motivational, stress or socialization information encoded in their frequency modulation pattern [6]. Likewise, Lusitanian toadfish males demonstrate their quality to females through their calling rate [7].

Acoustic signatures are also species-specific. Their evolution has a suggested role in the speciation process, as proposed in cichlids [8,9] or pipefishes [10]. In the latter, differences in the structure of sound producing apparatus including cranial bone morphology may explain the unique acoustic signatures of the feeding clicks produced by closely related species. Furthermore, within the piranha species *Serrasalmus marginatus*, red- and yellow-eyed morphs produce sounds with different amplitude features [11]. Genetic or hormonal differences could explain both the sound amplitude and the eye color, playing a role in communication. This is, to our knowledge, a rare case of within-species acoustic signature in fish. Thus, the evolutionary processes of acoustic signatures establishment within groups and in a speciation context remain largely unknown.

The teleost *Astyanax mexicanus* is widely used to investigate evolutionary genetic processes [12] and is a soniferous species [13]. Remarkably, acoustic communication has evolved between *Astyanax* surface-dwelling and blind cave-adapted morphotypes, which have diverged about 20.000 years ago [14,15]. *Astyanax c*avefish and surface fish share a repertoire of six sounds, but functionally the trigger, the use, and the meaning of one of these sounds, the "Sharp Click", has changed between surface fish and cavefish originating from La Cueva de El Pachón [13]. Therefore, acoustic communication seems to have evolved after the colonization of the subterranean habitat. The sound producing mechanisms in *Astyanax* are not known yet.

Remarkably, in northeastern Mexico there are more than 30 caves where cavefish populations live [16,17]. They are all blind and depigmented but show signs of ongoing genetic differentiation in different caves [18–21]. The ecological conditions in different caves can be very variable [16,17,22]. Moreover, the systematic exploration of sensory-driven behaviors in natural settings has revealed a significant amount of variations among caves. For example, a diversity of olfactory skills and responses to different odors exists among caves [23], and vibration attraction behavior mediated by the lateral line neuromasts is highly variable among different pools in a single cave [24]. In the same line, here we explored the variations and evolution of sound architecture among six cavefish populations that share the same sound repertoire [13]. We analyzed the acoustic features of "Clicks" and "Serial Clicks" produced by cavefishes, in their natural cave environment and we compared them to lab-raised fish originating from these same caves. We discovered cave-specific acoustic signatures.

## Materials and methods

### Field recordings

We recorded six different cavefish populations in natural settings (Sótano de Molino, Cueva de El Pachón, Cueva de los Sabinos, Sótano de la Tinaja, Cueva Chica, and Cueva del Río Subterráneo; abbreviated below by their location names) in March 2016, under the auspices of permit number SGPA/DGSV/02438/16 delivered by Secretaria de Medio Ambiente Y Recursos

Naturales of Mexico. Due to the topography and technical constraints encountered, recording setups, number of fish recorded and length of audio tracks varied between caves (see **S1 File** for Supplemental Methods). For example, the Molino cave being at the bottom of a 70 meter pit, we could not visit the cave 2 days in a row and obtain overnight acoustic recordings; or the water level being too low in the Tinaja perched pool, we could not set up a recording net and fish were recorded by simply hanging an hydrophone over their natural rocky pool.

Recordings were as described [13], using hydrophones (H2a-XLR, Aquarian Audio, Anacortes, USA; sensitivity: −180 dB, re 1 V/μPa, flat frequency response: ±4 dB, 20 Hz–4.5 kHz) connected to pre-amplifiers (ART Dual Pre USB) and recorders (Zoom H4n).

## Lab recordings

For four laboratory-bred cavefish line (originating from Molino, Pachón, Tinaja and Chica), one group of six adults was recorded in 25L tanks with foam overlaying glass walls, during 3hours after 1hour habituation. The recording chain was identical to [13]: hydrophones were connected to a pre-amplifier (OctaMic II, RME), linked to a Blackmagic Decklink 4K card. The distance between hydrophones and recorded fish was under the attenuation distance, estimated according to [13,25].

Fish were maintained in the laboratory of Dr. Clifford Tabin at Harvard Medical School under standard aquaculture conditions, IACUC approval #IS00001612-3. No euthanasia nor anesthesia procedure was used for this study.

## Sound extraction and analysis

Field sounds were extracted by ear and visual inspection of the sonograms with Audacity. Sonograms were magnified at a 3–4 seconds temporal window and then at a 0.2–1 second bin. Sound parameters were extracted using a R routine developed from the SeeWave R package [26]. Acoustic structure of Clicks and Serial Clicks were analyzed using 3 and 5 parameters (chosen after principal component analyses used to reduce the number of variables), respectively (see **S1 File**). For each cave the number of sounds analyzed (50–90 Clicks, 40–52 Serial Clicks) corresponded to minimum ten times the number of studied variables (see **S1 File**). Pulses were considered "Single" if they were of short duration (<20msec) and separated by >1sec interval from the next pulse.

Laboratory sounds were extracted and analysed with a supervised, machine learning-based automation tool developed in-house using Python 3.9 [27]; **S1 File**). Essentially, the recording was processed by a reference-signal matched filter [28]. The reference signal consisted of a Click randomly chosen by ear. The filter correlates the unique Click with the recording. The correlation maxima indicate the temporal positions of sounds matching the reference signal within the audio track. The process was placed under a supervised machine learning system. Namely, we used the matched filter on a 20min sequence, randomly chosen on the recording. We compared the results obtained with this process to by-ear examination of the same 20min. We refined the classification threshold parameters to reach/exceed 95% of true signals recognized, to the detriment of the total number of signals. Finally, we passed the matched filter on the whole recording. Detected clicks were isolated and their acoustic parameters extracted as above. We validated the automated tool by comparing the values of acoustic parameters after manual and automated extraction (**Table 1** and **S1 File**).

## Statistics

For each acoustic parameter of "Clicks" and "Serial Clicks", normality and homoscedasticity was assessed with respectively Kolmogorov-Smirnov tests and a Bartlett test. As normality and

**Table 1. Validation of the supervised machine-learning based tool.**

| SINGLE CLICKS | | | | |
|---|---|---|---|---|
| **Variable** | **Method** | **n** | **mean** | **±sem** |
| Dominant frequency (Hz) | manual | 48 | 2188 | 8.5 |
| | auto | 48 | 2059 | 9.4 |
| Duration (ms) | manual | 48 | 4.99 | 0.20 |
| | auto | 48 | 4.65 | 0.10 |
| SNR | manual | 48 | 7.55 | 1.23 |
| | auto | 48 | 11.11 | 0.15 |
| **SERIAL CLICKS** | | | | |
| **Variable** | **Method** | **n** | **mean** | **±sem** |
| Mean interpulse duration (ms) | manual | 47 | 15.40 | 2.98 |
| | auto | 45 | 23.84 | 3.32 |
| Mean pulse duration | manual | 47 | 4.18 | 0.19 |
| | auto | 45 | 6.64 | 0.07 |
| Pulse number | manual | 47 | 8.06 | 0.48 |
| | auto | 45 | 6.62 | 0.55 |
| Pulse rate | manual | 47 | 111.7 | 9.8 |
| | auto | 45 | 123.6 | 14.7 |
| Total duration (ms) | manual | 47 | 145 | 23.4 |
| | auto | 45 | 133 | 23.2 |

Legend: Comparison of cavefish sounds parameter values after manual or automated extraction and analysis on artificially built-in audio track containing a known quantity of Single or Serial Clicks that had been previously analyzed manually.

homoscedasticity were not systematically encountered, we proceeded to cave comparisons using Kruskal-Wallis tests followed with Dunn's *post hoc* tests. In figures, violin plots show the distribution and the median of samples. Statsoft Statistica 6, GraphPad Prism 9.4.1 and R 3.1.3 [29] were used for statistical analyses and graphical representations.

All sounds of the same type were also represented in the acoustic space formed by the two first principal axes of a principal component analysis (PCA) using respectively 3 and 5 variables for "Clicks" and "Serial Clicks". PCAs were followed by permuted discriminant function analysis (pDFA) to assess the quality of discrimination of PCAs (**S1 File**).

**A Source data file** containing the raw data presented and analyzed in this paper is available (**S1 Table**).

## Results

### Recording acoustic production of cavefish in natural caves

We performed acoustic recordings in six different natural caves. We chose Sótano de Molino, Cueva de El Pachón, Cueva de los Sabinos, Sótano de la Tinaja, Cueva Chica, and Cueva del Río Subterráneo because they are distributed throughout the three geographically distinct mountains ranges where *A. mexicanus* cavefish populations live (**Fig 1A**). Because of the specificities and practical limits encountered in each cave, the recording conditions, the number of fish recorded and the length of audio bands varied between caves (**Fig 1B** and **S1 File**). Available recordings were as follows: Molino (03/2017): 4 fish in openwork crate, 1h30; Pachón (03/2016): 10 fish in net, 11h; Los Sabinos (03/2017): 10 fish in net, 9h; Tinaja (03/2016), 25 fish in small natural pool, 1h30; Chica (03/2017), 20 fish in natural pond, 11h; Subterráneo (03/2016):

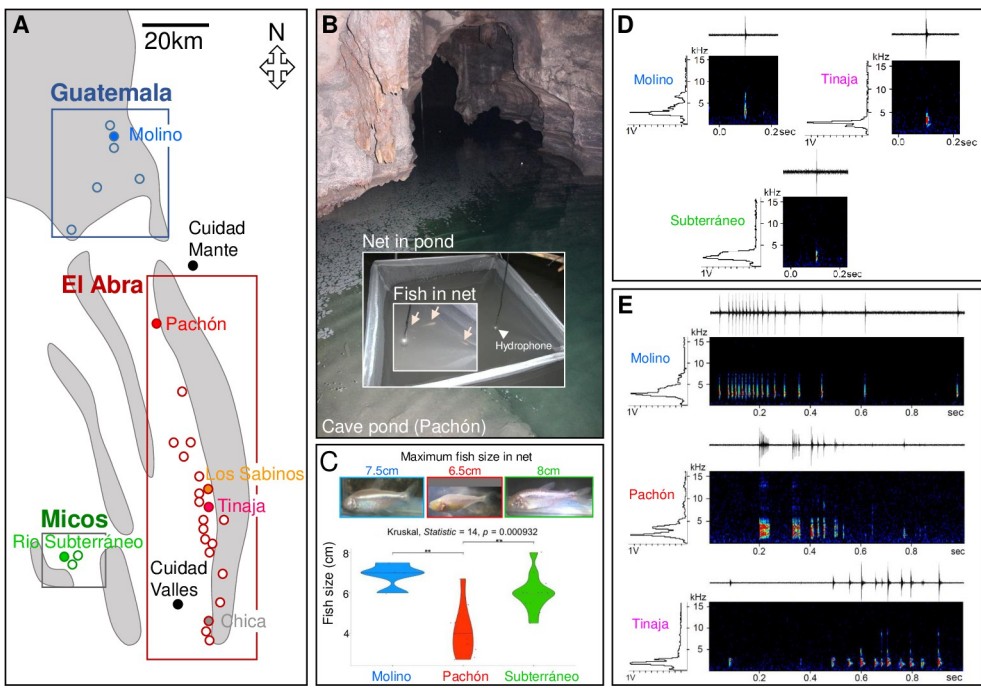

**Fig 1. Recording sounds in natural *A. mexicanus* caves. A:** Map showing the geographical localization of the 6 caves (color-coded) in the 3 mountain ranges (rectangles) where acoustic production was recorded, in deep caves in the field. **B:** Recording setup in natural settings, shown here as an example in the main pool of the Pachón cave. Cavefish were held in a large net and a hydrophone was hanged in the middle of the water column. **C:** Sizes and phenotypes of fish recorded in different caves. Photographs of the largest individuals present in the recording net are shown for each cave. **D, E:** Examples of sonograms of Clicks (D) and Serial Clicks (E) recorded in the wild, showing substantial level of variations across caves.

12 fish in net, 10h. In agreement with known genetic or ecological conditions previously reported across caves (e.g. [23,30,31], the phenotypes of the six recorded wild cavefish populations were variable in terms of size or level of troglomorphism. For example in 2016, the cavefish recorded in Pachón were 2.7 to 6.7 cm in length (including one juvenile) and were fully troglomorphic, while those recorded in Subterráneo were 4.5 to 8 cm in length and among them two had tiny eyes (**Fig 1C**) [31].

A first, global analysis of audio bands (total of 44h) was performed. Single and Serial Clicks were the most represented sounds produced by cavefish in their natural environment, while the other sounds of the repertoire [13] were rarer. We therefore focused our analyses on Clicks and Serial Clicks, the two sounds showing the largest frequency bandwidth (500–10,000Hz) (**Fig 1D and 1E**). We extracted them by ears (more than 100 sounds per cave and per sound type). For the best signals (between 50–90 Clicks per cave, between 40–52 Serial Clicks per cave), we extracted the acoustic parameters (23 for Clicks, 9 for Serial Clicks) and we compared the most relevant (the less correlated, *i.e.* 3 parameters for Clicks and 5 parameters for Serial Clicks) between caves (**S1 Table**).

## Clicks and Serial Clicks vary between different wild cavefish populations

Concerning Single Clicks, the sound duration, the dominant frequency, and the signal to noise ratio (SNR) all varied significantly among the six caves (**Fig 2A**; Kruskal-Wallis statistics, p = 2.48e-18, p = 0.00000385 and p = 1.52e-13, respectively). Clicks were longest in Tinaja, most high-pitched in Molino. The SNR was high in Chica and low in Tinaja. Of note, the

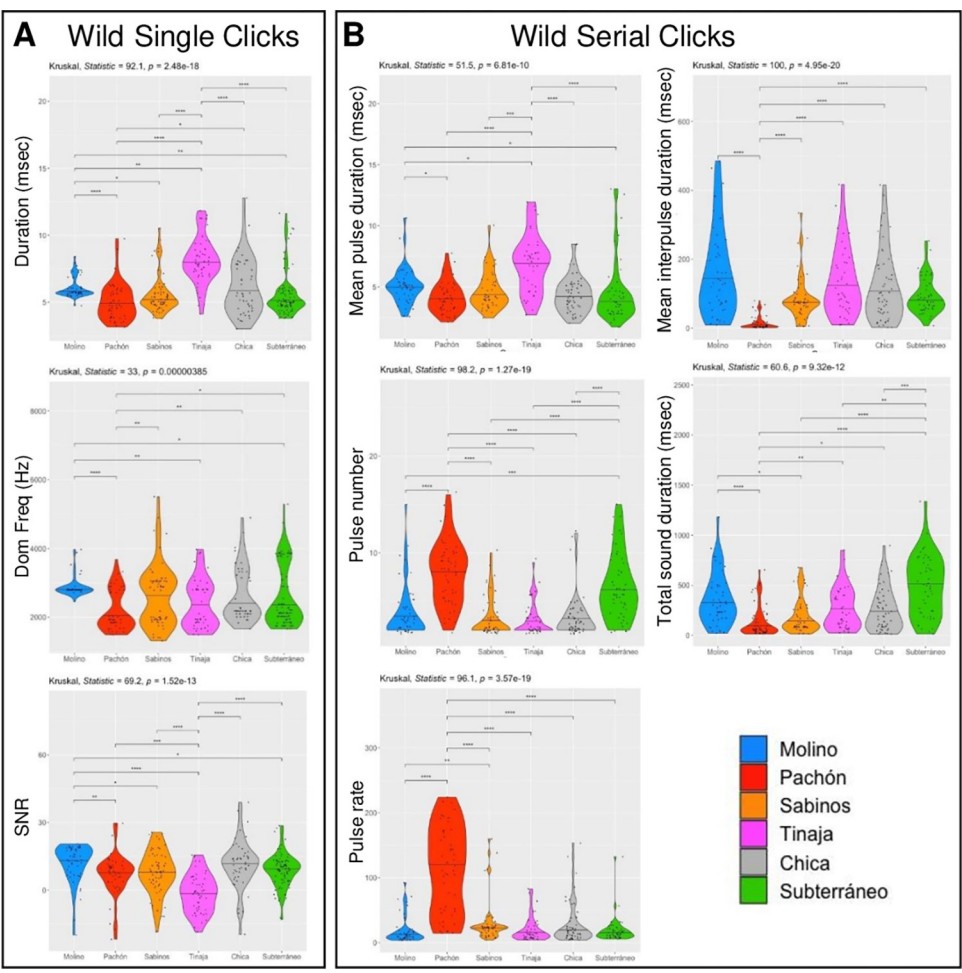

**Fig 2. Acoustic parameters of Clicks and Serial Clicks from wild *A. mexicanus* recorded in natural caves. A, B:** Univariate analysis of acoustic parameters in different caves (color-coded, see inset) for Clicks (A) and Serial Clicks (B). P values from Kruskal-Wallis statistics are indicated.

duration and SNR variances were highest in Chica, suggesting less homogeneity in the sound production, possibly related to the hybrid genetic background of the fish in this cave that is due to hybridization with surface fish [17,32]. For Serial Clicks, we focused on the parameters related to their multi-pulse nature and their sound envelope, which were also all variable among cave populations (**Fig 2B**; Kruskal-Wallis statistics, p = 6.81e-10 for pulse duration, p = 1.27e-19 for pulse number, p = 3.57e-19 for pulse rate, p = 4.95e-20 for interpulse duration and p = 9.32e-12 for total sound duration). Consistent with Single Clicks, the pulse duration of Serial Clicks were longer in Tinaja. In Pachón, the pulse rate was impressively high, about ten times higher on average than in the other caves, accompanied by a twice-higher number of pulses, a shorter inter-pulse duration and a shorter total duration of the sound. Other significant features were high pulse numbers and long total sound durations in Subterráneo, and long interpulses in Molino. Together, these data strongly suggest that cavefish sounds differ among caves. However, no apparent correlation or order of sound parameters emerge relative to fish phenotypes (e.g., small Pachón cavefish can produce high pulse rates) or cave phylogeography (e.g., Pachón and Subterráneo cavefish share a high number of pulses in their Serial Clicks).

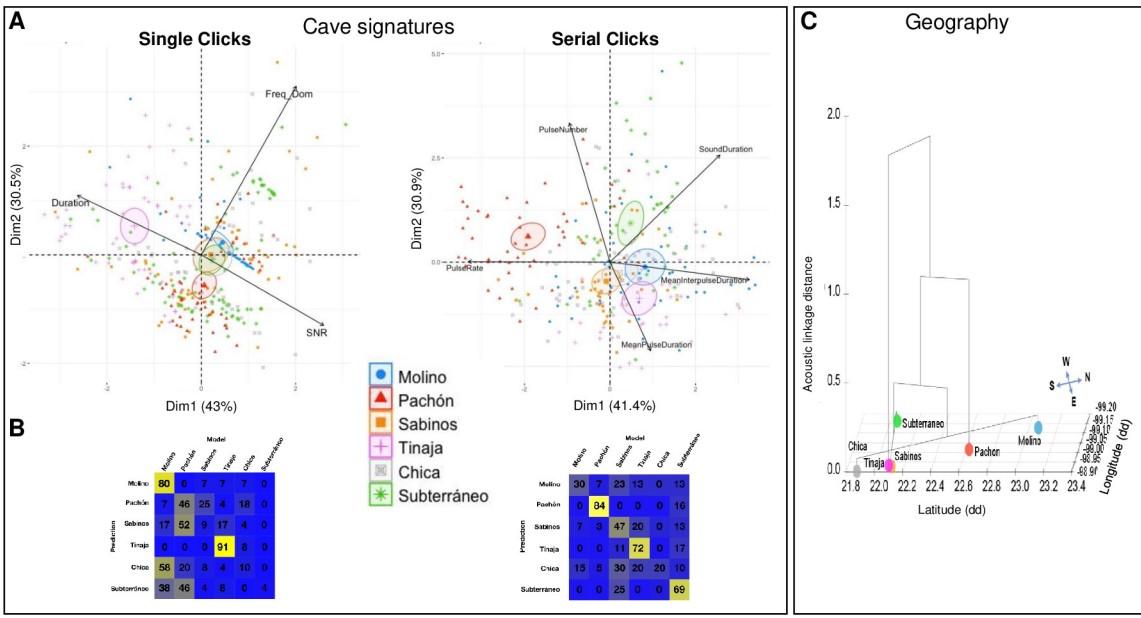

**Fig 3. Clustering of acoustic parameters of Clicks and Serial Clicks from wild *A. mexicanus* recorded in natural caves. A:** Cave sounds clustering in acoustic space using PCA. Left, Single Clicks; Right, Serial Clicks. The cave color code is indicated, 95% confidence ellipses are shown. **B:** Cave sounds reclassification with pDFA. **C:** Acoustic-phylogeographic relationship. Hierarchical clustering of acoustic parameters of Clicks (height axis) was projected onto a map of geographical coordinates of caves (NSWE in x and y-axes).

We next performed principal component analyses (PCA) in the acoustic space to evaluate the possibility of cave signatures (**Fig 3A**). For both Clicks and Serial Clicks, 95% confidence ellipses around centroids on the PCA showed little overlap and were mostly well separated (except for Chica, see above). Moreover, pDFA (permutated Discriminant Function Analysis) generated confusion matrixes with good scores of correctly reclassified sounds, confirming that the sounds produced in each cave could carry a specific acoustic signature (**Fig 3B**; p = 0.001). The Single Clicks of Molino (80% reclassification), Pachón (46%) and Tinaja (91%) were particularly distinctive, and the Serial Clicks from all caves except Chica showed significant reclassification scores (**Fig 3B**). Finally, using a hierarchical clustering tree onto geographical coordinates of caves in the PCA, we found some grouping of caves that did not fit with phylogeography (**Fig 3C**, Single Clicks). The apparent closeness of Subterráneo (Micos group) and Sabinos (El Abra group), or Molino (Guatemala group) and Chica (El Abra group) rather suggested independent evolution of sound production in these different caves. In sum, our analyses suggest the evolution of an "accent" in the different cavefish populations.

## Clicks and Serial Clicks vary between lab-raised cavefish populations of different origins

To strengthen and confirm the results obtained in "uncontrolled" field conditions, we took advantage of having access to four of these six cavefish strains studied above, maintained in laboratory facility. This allowed testing whether local environment can influence sound architecture in the wild, and whether acoustic signatures persist or even further evolve in long-term, laboratory-bred cavefish. We recorded groups of six individuals of the Molino, Pachón, Tinaja and Chica cavefish laboratory lines, in acoustic-proof tanks in the laboratory. Audio bands were processed by an original automated tool developed in-house (see **Methods**) allowing automated sound extraction and analysis. There, the automatic method allowed extracting

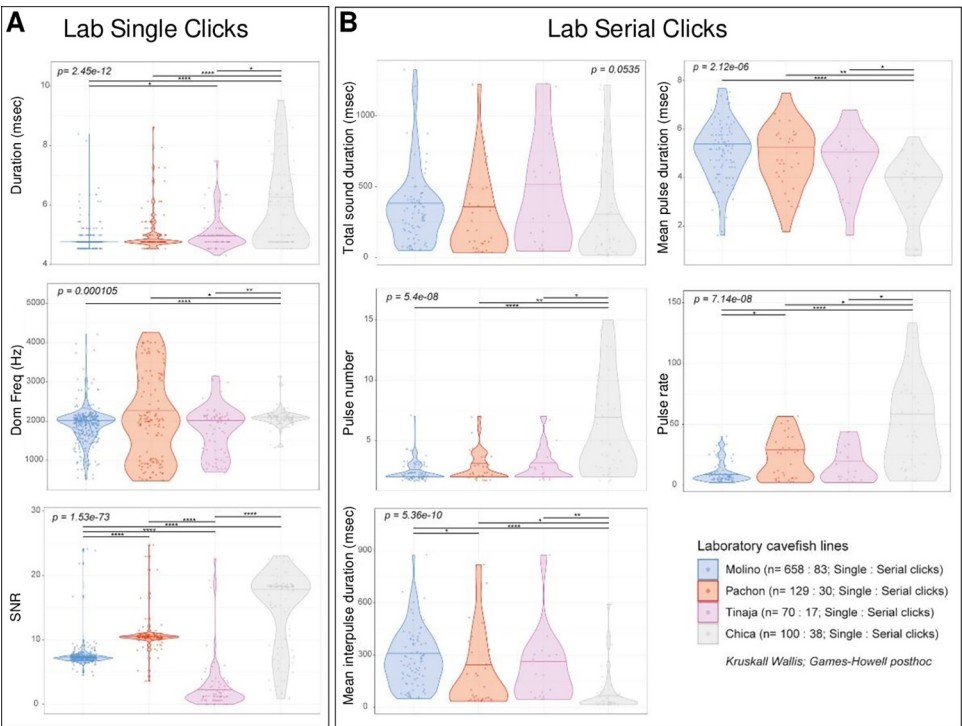

**Fig 4. Acoustic parameters of Clicks and Serial Clicks from lab-raised *A. mexicanus* lines, recorded in the lab. A, B:** Univariate analysis of acoustic parameters in laboratory-raised lines originating from different caves (color-coded, see inset) for Single Clicks (A) and Serial Clicks (B). P values from Kruskal-Wallis statistics are indicated.

more sounds per cave strain, i.e., for example we obtained 614 Single Clicks for the four cave strains—whereas we had a total of 200 sounds in field conditions. We compared the results obtained for the complete dataset (n = 614) and for a set of 50 randomly chosen sounds per cave (thus n = 200). We obtained very similar results (see **S1 File**), which further comforted the accuracy and the representativeness of the analyzed sample size for the field data.

Again, in the laboratory-bred cavefish, significant differences in Single Clicks structure existed between lines originating from different caves (**Fig 4A**; Kruskal-Wallis statistics, p = 2.45e-12 for duration, p = 0.000106 for dominant frequency, and p = 1.53e-73 for SNR). Some were consistent with field recordings (e.g., long duration in Chica and Tinaja; low SNR in Tinaja), but some were not (e.g, lab-raised Molino were not high-pitched). The same applied to Serial Clicks (**Fig 4B**; Kruskal-Wallis statistics, p = 0.053 for total sound duration, p = 5.4e-08 for pulse number, p = 5.36e-10 for interpulse duration, p = 2.12e-06 for pulse duration and p = 7.14e-08 for pulse rate). Some Serial Clicks features were shared with wild-recorded sounds (e.g., long interpulse in Molino) and some were not (e.g., lab-raised Pachón Serial Clicks were not particularly distinctive).

PCA on lab-recorded sounds confirmed that Single Clicks and, to a lesser extent, Serial Clicks which are rarer in tank recordings and therefore less numerous in the dataset, segregate on the acoustic space (**Fig 5A**) and show high and significant scores of reclassification post-permutations (**Fig 5B**). Of note, in contrast to wild Chica sounds, both Clicks and Serial Clicks from the lab-raised Chica line were well isolated in a separated cluster. A possible explanation could be that they were bred for decades in captivity without renewed gene flow and exchange with surface population–a phenomenon that occurs in the natural Cueva Chica.

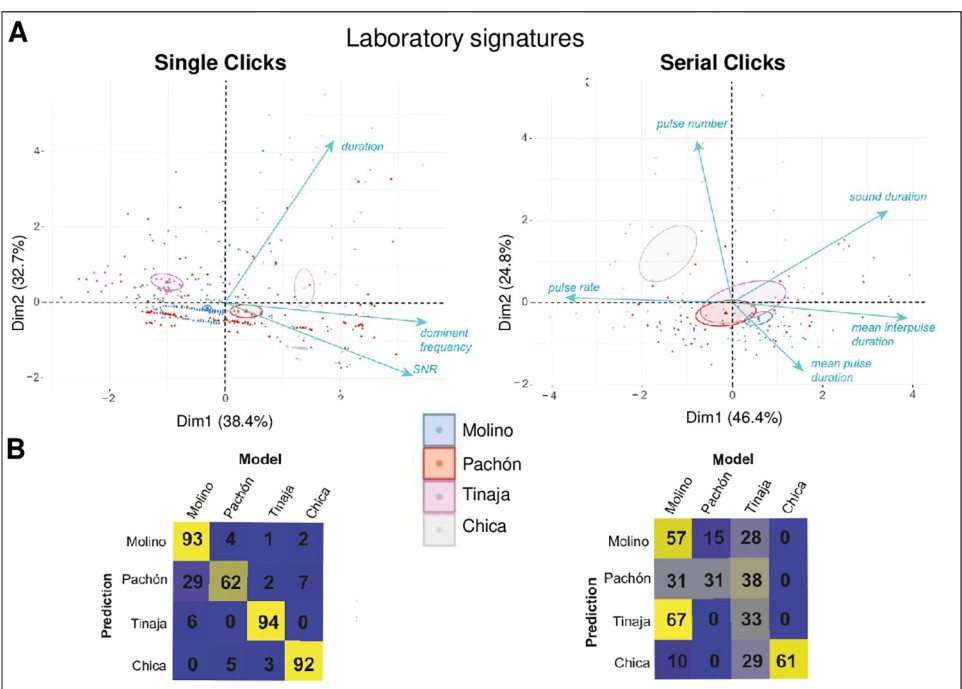

**Fig 5. Clustering of acoustic parameters of Clicks and Serial Clicks from lab-raised *A. mexicanus* lines, recorded in the lab. A**: Laboratory-bred cavefish lines sounds clustering in acoustic space using PCA. Left, Single Clicks; Right, Serial Clicks. **B**: Laboratory sounds reclassification with pDFA.

## Comparing acoustic signatures between wild and lab-raised cavefish from different origins

Finally, we sought to test the consistency of the proposed acoustic signatures by comparing sounds from wild and lab-raised cavefish populations. **Fig 6** shows the results of analyses for Single Clicks only (Serial Clicks are less numerous in the dataset for some caves, making the comparisons more difficult).

First, we performed a PCA including 8 groups of sounds: Clicks from wild Chica, Molino, Pachón and Tinaja cavefish, and Clicks from lab-raised Chica, Molino, Pachón and Tinaja lines (**Fig 6A**). Some ellipses (95% confidence around centroids) did overlap; some did not. Namely, a drift between wild and lab-bred Tinaja Clicks (pink shades) was observed in the first dimension of the PCA that mainly corresponds to sound duration, but there was no difference in the other dimension of the PCA. Clicks from wild and lab-bred Molino fish (blue shades) separated on the two dimensions. However, the Clicks produced by wild and lab-bred Pachón (red shades) and Chica (grey shades) showed overlapping confidence ellipses, suggesting that they are not differentiated. Kruskal-Wallis statistics on the comparison of the 8 groups for the first axis of the PCA (which mainly represents the sound duration) confirmed these observations (**Fig 6C**). Overall, this analysis confirms that Single Clicks produced by cavefish from different backgrounds segregate in the acoustic space, with a tendency to grouping clusters when they originate from the same cave.

Second, we performed a PCA with 4 groups of sounds, by artificially pooling Clicks recorded from wild and lab-bred cavefish originating from a given cave (**Fig 6B**). There, the galaxies representing the Clicks produced by the Chica, Molino, Pachón and Tinaja cavefish, disregarding their wild or lab-bred origin, were well separated and showed non-overlapping confidences ellipses. This was also supported by Kruskal-Wallis statistics on the comparison of

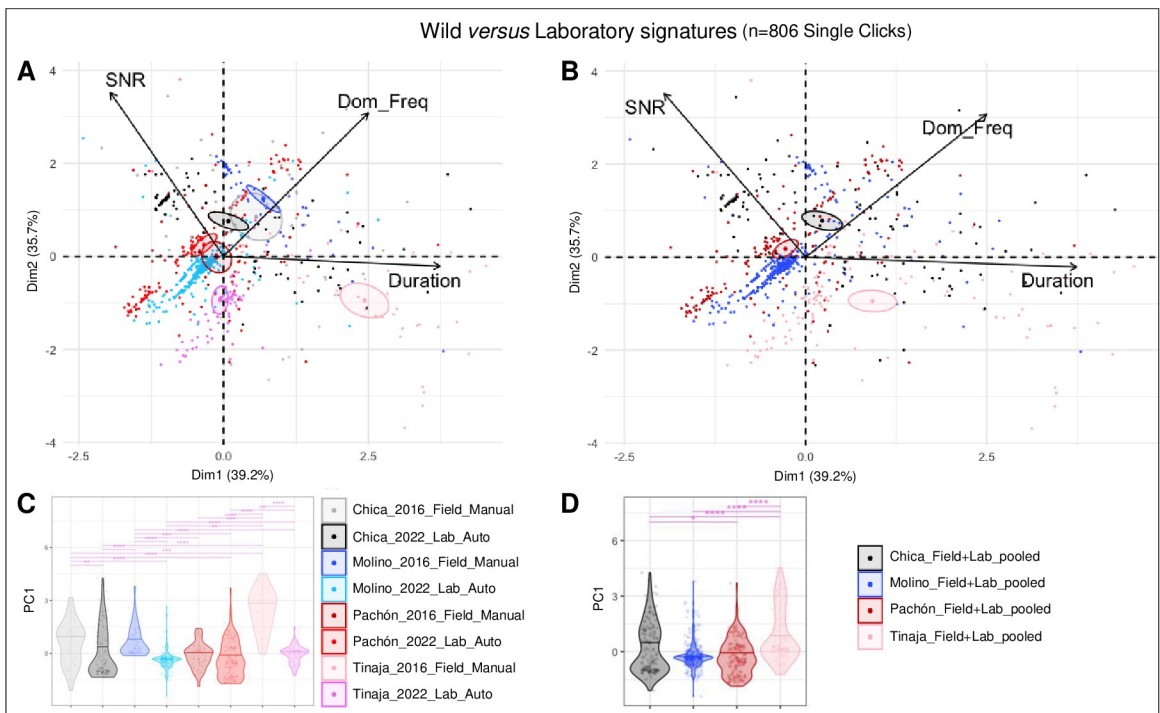

**Fig 6. Comparison of acoustic parameters of Clicks from wild and lab-raised *A. mexicanus* originating from different caves. A, C:** Clustering of 8 groups of sounds (4 wild cave recordings, 4 lab lines recordings) using PCA (A). Violin plots showing the comparison of the first axis (PC1) of the PCA for the 8 groups, with Kruskall-Wallis statistics (C). The color codes are indicated. **B, D:** Clustering of 4 groups of sounds (4 different caves, wild cave-recorded Clicks and lab lines recorded Clicks were pooled) using PCA (B). Violin plots showing the comparison of the first axis of the PCA for the 4 groups, with Kruskall-Wallis statistics (D). The color codes are indicated.

the 4 groups for the first axis of the PCA (**Fig 6D**). This analysis further confirms the existence of acoustic signatures related to the location where cavefish originate: wild and lab-bred cavefish from Chica, Molino, Pachón and Tinaja background did not live and were not recorded in the same environment, yet a cave-of-origin specific signature emerges.

## Discussion

The existence of acoustic signatures in recently evolved populations of cavefish was unexpected. Our findings provide a promising model to study the proximal and distal mechanisms for the evolution of acoustic communication in a species on its way to diversification and speciation [33]—even though *Astyanax* morphs still belong to the same species and show little genetic differentiation. The most likely origins for observed differences between cave populations could include plasticity in response to specific local biotic and abiotic ecological conditions [16,23], or the independent and subtle morphological evolution of facial and jaw bones in relation with the loss of eyes in cavefishes [34]. As signatures were also found in laboratory-bred animals, we favor the latter hypothesis. This would imply that the developmental evolution of the cavefish head not only affects the visual, olfactory and mechano-sensory but also the acoustic facet of their communication modalities, in a pleiotropic manner.

Most interestingly, even though the sound architecture in wild- and lab-recorded cavefish of the same origin showed some variations, both wild fish and laboratory lines did segregate in the acoustic space. Moreover, analyses comparing sounds produced by wild and lab-raised cavefish from the same origin confirms the existence of such signatures, irrespective of the fact

they had very different life experiences and had been recorded in very different environmental conditions. Such preservation in laboratory settings suggests that the cave-specific acoustic signatures have a genetic basis. Our findings also suggest that cave-specific signatures could persist and further evolve in captivity, and that cavefish accents may be labile and not permanently fixed. Sound architecture does not seem to evolve in a predictable manner according to geography or local biotic and abiotic conditions, and may rather reflect a degree of isolation of cavefish populations. Importantly, these populations are small, in the order of a few hundreds to a few thousands of individuals [35]. Therefore, we propose that the evolution of such acoustic signatures would be neutral and occur by drift, progressively leading to the differentiation of local accents that may ultimately prevent interbreeding and contribute to speciation.

The function(s) of acoustic communication among cavefish are mostly unknown. Previously, we have shown that Sharp Clicks, one of the six sounds *Astyanax* can produce, are emitted during chemosensory-driven foraging [13]. The meaning or significance of other sounds including Clicks is unknown, and it will be extremely interesting to test their potential function in foraging, or breeding or any other social interaction or behavior. In the subterranean environment too, the soft chirps of naked mole-rats encode individual identity as well as colony identity and they are culturally transmitted as colony vocal dialects, carrying information about group membership [36]. Although cavefish are supposed to be asocial, they are capable of social-like interactions in familiar environments [37]. The cave-specific acoustic signatures, or accent, we have discovered may well participate in such sociality when thriving in their natural caves.

## Supporting information

**S1 File. A supplemental methods file.**
(PDF)

**S1 Table. A Source data file (S1 Table) containing the raw data presented and analyzed in this paper.**
(XLSX)

## Acknowledgments

We thank Luis Espinasa, Julien Fumey, Stéphane Père and all members of the Rétaux's lab for their helpful spirit in the field, Patricia Ornelas-Garcia for obtaining shared fieldwork permits, Brian Martineau and the aquatic facility of the Tabin lab for animal care and Joshua Gross for his donation of the Chica cavefish. We thank Clifford Tabin for help and support.

## Author Contributions

**Conceptualization:** Carole Hyacinthe, Joël Attia, Sylvie Rétaux.

**Data curation:** Carole Hyacinthe, Joël Attia, Lény Lego, Sylvie Rétaux.

**Formal analysis:** Carole Hyacinthe, Joël Attia, Elisa Schutz.

**Funding acquisition:** Joël Attia, Sylvie Rétaux.

**Investigation:** Carole Hyacinthe, Joël Attia, Elisa Schutz, Didier Casane, Sylvie Rétaux.

**Methodology:** Carole Hyacinthe, Joël Attia, Elisa Schutz, Lény Lego, Sylvie Rétaux.

**Project administration:** Sylvie Rétaux.

**Resources:** Sylvie Rétaux.

**Software:** Joël Attia, Lény Lego.

**Supervision:** Joël Attia, Sylvie Rétaux.

**Validation:** Carole Hyacinthe, Joël Attia, Sylvie Rétaux.

**Visualization:** Carole Hyacinthe, Joël Attia, Sylvie Rétaux.

**Writing – original draft:** Sylvie Rétaux.

**Writing – review & editing:** Carole Hyacinthe, Joël Attia, Sylvie Rétaux.

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
