## [Decision Letter · Decision Letter 0]

13 Jul 2023

PONE-D-23-18782Acoustic signatures in Mexican cavefish populations inhabiting different cavesPLOS ONE

Dear Dr. Rétaux,

Thank you for submitting your manuscript to PLOS ONE. After careful consideration, we feel that it has merit but does not fully meet PLOS ONE’s publication criteria as it currently stands. Therefore, we invite you to submit a revised version of the manuscript that addresses the points raised during the review process.

We look forward to receiving your revised manuscript.

Kind regards,

Hector Escriva, PhD

Academic Editor

PLOS ONE

Journal Requirements:

"Work supported by a Lidex Neuro-Saclay collaborative grant to SR and JA, an Equipe FRM grant (DEQ20150331745) to SR, and an Ecos-Nord exchange Program (M15A03) to SR and Patricia Ornelas-Garcia. Fondation des Treilles prize fellowship to CH.

We thank Luis Espinasa, Julien Fumey, Stéphane Père and all members of the Rétaux’s lab for their helpful spirit in the field, Patricia Ornelas-Garcia for obtaining shared fieldwork permits, Brian Martineau and the aquatic facility of the Tabin lab for animal care and Joshua Gross for his donation of the Chica cavefish. We thank Clifford Tabin for help and support. "

'Work supported by:

a Lidex Neuro-Saclay collaborative grant to SR and JA (no website), 

an Equipe FRM grant (DEQ20150331745) to SR (https://www.frm.org/), 

an Ecos-Nord exchange Program (M15A03) to SR and Patricia Ornelas-Garcia (https://www.univ-spn.fr/ecos-nord/).

a Fondation des Treilles prize fellowship to CH (https://www.les-treilles.com/).

5. We note that Figure 1a in your submission contain [map/satellite] images which may be copyrighted. All PLOS content is published under the Creative Commons Attribution License (CC BY 4.0), which means that the manuscript, images, and Supporting Information files will be freely available online, and any third party is permitted to access, download, copy, distribute, and use these materials in any way, even commercially, with proper attribution. For these reasons, we cannot publish previously copyrighted maps or satellite images created using proprietary data, such as Google software (Google Maps, Street View, and Earth). For more information, see our copyright guidelines: http://journals.plos.org/plosone/s/licenses-and-copyright.

a. You may seek permission from the original copyright holder of Figure 1a to publish the content specifically under the CC BY 4.0 license.  

6. Please include a copy of Table 1 which you refer to in your text on page 6.

Reviewers' comments:

Reviewer's Responses to Questions

**Comments to the Author**

1. Is the manuscript technically sound, and do the data support the conclusions?

Reviewer #1: Yes

Reviewer #2: Yes

2. Has the statistical analysis been performed appropriately and rigorously? 

Reviewer #1: Yes

Reviewer #2: Yes

3. Have the authors made all data underlying the findings in their manuscript fully available?

Reviewer #1: Yes

Reviewer #2: Yes

4. Is the manuscript presented in an intelligible fashion and written in standard English?

Reviewer #1: Yes

Reviewer #2: Yes

5. Review Comments to the Author

Reviewer #1: The manuscript: “Acoustic signatures in Mexican cavefish populations inhabiting different caves” by Hyacinthe et al. presents a study on the acoustic communication of cavefish populations in different natural caves. The researchers conducted acoustic recordings in six caves and analyzed the sound production of the cavefish populations. They aimed to determine if there were distinct acoustic signatures or "accents" associated with different cave populations and if these signatures were genetically maintained in the laboratory.

The study recorded various types of sounds produced by the cavefish, with a focus on Clicks and Serial Clicks, which showed the largest frequency bandwidth. They extracted acoustic parameters from these sounds and compared them between caves to identify any significant variations.

The results of the analysis revealed that both Clicks and Serial Clicks exhibited significant differences among the cavefish populations in terms of sound duration, dominant frequency, and signal-to-noise ratio. Excitingly, these differences were observed both in the recordings from the wild populations and the laboratory-bred populations, despite differences in their living environments and life experiences, indicating that it is a robust genetic trait.

The findings of this study have important implications for understanding the evolution of acoustic communication and potential speciation in cavefish populations. The presence of distinct acoustic signatures among different cave populations indicates the possibility of acoustic divergence and the development of local accents over time. These acoustic differences could contribute to the reproductive isolation and speciation of cavefish populations which demonstrates the potential for using acoustic analysis as a tool for studying the evolution and speciation of species.

Overall, the paper is well-written and provides detailed information about the methods used, the results obtained, and their implications. The findings contribute to the understanding of acoustic communication in cavefish populations and open up avenues for further research in this field.

I have only minor comments:

The abstract does not do justice to the exciting findings of the paper. I recommend revising it to emphasize the identification of distinct cave-specific genetic traits and their preservation within the laboratory setting, indicative of a robust genetic basis.

Given that the data does not offer conclusive evidence supporting or refuting genetic drift or selection, I propose excluding the discussion on this topic from both the abstract and the main text.

It would have been nice to have surface fish included in the study, however, I understand if this is beyond the scope of this study.

Reviewer #2: This interesting manuscript examines the acoustical signatures of natural cavefish from the El Abra region of Mexico. Intriguingly, these fish were discovered to produce sounds that may be important for communication among members of the same cave locality. Herein, the authors provide a diverse set of analyses of these acoustic signatures and find that non-captive (i.e., "native") cave populations have specific acoustic signatures, and these signatures are likely subject to drift as their principal evolutionary mechanism explaining differences across cave populations. Overall, this is a very interesting and well-conducted studies that is appropriate for publication. The manuscript is well written and presented, and the structure is nicely organized. Below, I provide my comments for the authors to consider - the fact that many are editorial/discretionary speaks to the quality of the submitted manuscript.

Comments:

1. The authors summarize broadly the source of acoustic sounds in other teleost species - is the source of sound generation in cave (or surface) morphs known? (e.g., swim bladder v. stridulation of cranial bones?)

2. The authors reference 'practical limits in each cave' and 'recording conditions'. Can you clarify precisely how these may impacted the collected data for the reader?

3. Please cite the described hybrid origin of Chica fish, as referenced in L190 and L254.

4. This interesting report focuses largely on the production of sound, but less on the reception of sounds by conspecific members of the same locality. Can the authors speak to a response phenotype, that could potentially be examined, that would connect the production of within-cave sounds to reception by other members within the cave? This would seem to ratify both the importance of sounds for a phenotypic outcome (e.g., breeding?), and reinforce the assertion that sound production/response is specific to individual caves.

Minor, discretionary suggestions:

1. L79 - "northeastern" should not be capitalized

2. L80 - are all the cave-populations blind? My understanding was that Caballo Moro has some purportedly sighted fish.

3. L130 - "...consisted of a click..."

4. L138: "clicks" should be lowercase.

5. The sentence ending on L139 ends abruptly.

6. L193: 'pule' should be 'pulse'

6. PLOS authors have the option to publish the peer review history of their article (what does this mean?). If published, this will include your full peer review and any attached files.

Reviewer #1: No

Reviewer #2: No

---

## [Author Response · Author response to Decision Letter 0]

19 Jul 2023

Detailed responses to reviewers and journal's requirement are listed in the "response to reviewers" file uploaded

---

## [Editor Report · Decision Letter 1]

21 Jul 2023

Acoustic signatures in Mexican cavefish populations inhabiting different caves

PONE-D-23-18782R1

Dear Dr. Rétaux,

We’re pleased to inform you that your manuscript has been judged scientifically suitable for publication and will be formally accepted for publication once it meets all outstanding technical requirements.

Kind regards,

Hector Escriva, PhD

Academic Editor

PLOS ONE
---

## [Editor Report · Acceptance letter]

26 Jul 2023

PONE-D-23-18782R1 

Acoustic signatures in Mexican cavefish populations inhabiting different caves 

Dear Dr. Rétaux:

I'm pleased to inform you that your manuscript has been deemed suitable for publication in PLOS ONE. Congratulations! Your manuscript is now with our production department. 

Kind regards, 

on behalf of

Dr. Hector Escriva 

Academic Editor

PLOS ONE